# OpenReview forum: "Vulnerabilities Mitigation for Safety-Aligned Language Models via Debiasing"
_ICLR.cc/2025/Conference — Submitted to ICLR 2025_

### Official Review · Reviewer_c1NK · 2024-10-23

**Soundness:** 2
**Presentation:** 3
**Contribution:** 2
**Rating:** 6
**Confidence:** 4

**Summary:**

This paper first reveals the vulnerability of safety-aligned models in certain categories and then identifies the potential bias of generating negative tokens induced by safety alignment. To address such an issue, the authors propose TSDI, a token-level safety debiased inference method, to mitigate such bias during the generation phase. The experimental results show that the proposed method can improve the safety-helpfulness of the Pareto front with better safety levels while preserving the helpfulness of the models.

**Strengths:**

1. Timely and vital problem.
2. A new debiasing approach to improving the trade-off between the safety and helpfulness of safety-aligned LLMs.

**Weaknesses:**

This paper tackles the vulnerability of safety-aligned language models from a debiasing perspective, which is interesting and deserves more exploration. However, there are several concerns regarding the problem formulation, methodology and evaluation of this paper, which makes the rigor of the methodology and the validity of the conclusion questionable. Please see the detailed comment below.

1. The problem formulation of "vulnerability of safety-aligned models" is unclear. In Section 2.2, the authors state, "Our work highlights a different type of vulnerability, in which a model, despite being safety-aligned and deemed safe overall, generates harmful responses on specific topics." To me, it is ambiguous how this vulnerability differs from the conventional safety issues in LLMs, as LLM safety is a border concept that covers various topics.

2. Throughout the paper, the authors claimed several times that the proposed method improves the safety levels of the subject LLMs across "ALL" safety categories. Conversely, the experiments are conducted mainly in the category of adult content; thence, it is not clear to what extent TSDI is still effective from other safety perspectives and the potential interaction among those safety categories.

3. Many concepts and implementation details are not fully explained. For instance, the paper does not provide a clear definition of helpfulness as well as the coreespond metrics (e.g., helpfulness win rate). Such a lack of explanations makes the paper somehow hard to follow.

4. In Section 3.1, the safety score is computed based on the percentage of safe responses classified by two safety evaluators. How do the percentages from the evaluators get fused to the final score? Since the evaluators themself can possibly exhibit certain degrees of bias on specific safety categories, are any actions taken to mitigate such an issue?

5. This paper subtracts the bias estimated from randomly generated prompts to reduce the unintentional biases of negative tokens; however, it is not intuitive to eliminate the bias from output logits in all cases. In particular, the biases of negative tokens in output logits may vary according to different safety scenarios; that is, the degree of the bias can change based on unharmful prompts and harmful prompts with different types of safety categories. For instance, it is reasonable that for specific harmful inputs, the models exhibit increased logits for negative tokens. Therefore, simply subtracting the bias measured from randomly generated prompts does not seem like a rational way that applies to all cases.

6. The evaluation metrics introduced in Section 5.1 are confusing. The authors used compliance rate and helpfulness win rate to measure the helpfulness improvement. Specifically, the compliance rate assesses the helpfulness based on particular tokens and GPT-4 is used to measure the "actual helpfulness." What does "actual helpfulness" mean here? Has the compliance rate considered cases where an unharmful prompt is refused to be answered?

**Questions:**

1. How does the "different type of vulnerability" mentioned by the authors differ from the conventional LLM safety vulnerabilities?

2. Is there evidence showing that TSDI is effective in "all safety categories," as the authors claim (I do not observe a significant improvement from Figure 1)?

---

> ### Author Response · Authors · 2024-11-19
> **Authors' replies to Reviewer c1NK (1/2)**
>
> We deeply appreciate the reviewer’s encouraging comments. The thoughtful comments of Reviewer c1NK are valuable and help us improve the manuscript's quality. We will answer the comments in Weaknesses and Questions.
>
> > W1: The problem formulation of "vulnerability of safety-aligned models" is unclear. In Section 2.2, the authors state, "Our work highlights a different type of vulnerability, in which a model, despite being safety-aligned and deemed safe overall, generates harmful responses on specific topics." To me, it is ambiguous how this vulnerability differs from the conventional safety issues in LLMs, as LLM safety is a border concept that covers various topics.
>
> A: Please see the response of Question 1.
>
> > W2: Throughout the paper, the authors claimed several times that the proposed method improves the safety levels of the subject LLMs across "ALL" safety categories. Conversely, the experiments are conducted mainly in the category of adult content; thence, it is not clear to what extent TSDI is still effective from other safety perspectives and the potential interaction among those safety categories.
>
> A: We deeply thank the reviewer for the valuable comment. In Figure 1 of the original submission, we evaluated the safety levels of TSDI across all categories to demonstrate its high safety level. However, we acknowledge that additional evidence would strengthen our claim. Therefore, to provide more comprehensive support for the effectiveness of our proposed method in various categories, we conducted new experiments and now present new figures in Section 5 of our revised paper. Specifically, we additionally provide new experiment results on two categories: 1) Trade and Compliance category and 2) Security Threats in addition to adult content. We would like the reviewer to go through our experiment section, which we believe resolves the reviewer's concerns.
>
> > W3: Many concepts and implementation details are not fully explained. For instance, the paper does not provide a clear definition of helpfulness as well as the coreespond metrics (e.g., helpfulness win rate). Such a lack of explanations makes the paper somehow hard to follow.
>
> A: Thank you for your helpful comment. We have provided a better explanation of the helpfulness metrics used in the paper. Though we consider our revised version more readable than the initial version, we would be grateful if there are still unclear points.
>
> > W4: In Section 3.1, the safety score is computed based on the percentage of safe responses classified by two safety evaluators. How do the percentages from the evaluators get fused to the final score? Since the evaluators themself can possibly exhibit certain degrees of bias on specific safety categories, are any actions taken to mitigate such an issue?
>
> A: Since different evaluators have different evaluation criteria, we used multiple evaluators to prevent unfair or biased experimental results. In practice, a single safety evaluator will typically be chosen based on the users' demands; hence, it is out of scope to fuse the scores of the multiple evaluators. That said, we agree with the reviewer that it would be a good idea to combine the safety scores of multiple evaluators.

---

> > ### Author Response · Authors · 2024-11-19
> > **Authors' replies to Reviewer c1NK (2/2)**
> >
> > > W5: This paper subtracts the bias estimated from randomly generated prompts to reduce the unintentional biases of negative tokens; however, it is not intuitive to eliminate the bias from output logits in all cases. In particular, the biases of negative tokens in output logits may vary according to different safety scenarios; that is, the degree of the bias can change based on unharmful prompts and harmful prompts with different types of safety categories. For instance, it is reasonable that for specific harmful inputs, the models exhibit increased logits for negative tokens. Therefore, simply subtracting the bias measured from randomly generated prompts does not seem like a rational way that applies to all cases.
> >
> > A: As the reviewer mentions, there may exist biases that are specific to prompts or categories. However, it is extremely difficult to identify such biases, and we also need to identify which category each prompt belongs to during the inference of LLMs. Our proposed TSDI is not specific to the adult content category and we consider it to a general algorithm. As evidence, we added new experiments and presented new figures in Section 5 of our revised paper. Specifically, we conducted the same experiments for 1) Trade and Compliance category and 2) Security Threats in addition to 3) adult content. We would like the reviewer to go through our experiment section, which we believe resolves the reviewer's concerns. Although the reviewers may not consider our TSDI intuitive, we believe our TSDI works as a general algorithm, which is supported by our new empirical experiments.
> >
> > > W6: The evaluation metrics introduced in Section 5.1 are confusing. The authors used compliance rate and helpfulness win rate to measure the helpfulness improvement. Specifically, the compliance rate assesses the helpfulness based on particular tokens and GPT-4 is used to measure the "actual helpfulness." What does "actual helpfulness" mean here? Has the compliance rate considered cases where an unharmful prompt is refused to be answered?
> >
> > A: We included two helpfulness metrics: compliance rate and helpfulness win rate because they evaluate different aspects of helpfulness in the response. The compliance rate assesses whether the models refuse to answer an unharmful prompt by checking the initial tokens. On the other hand, the helpfulness win rate focuses more on the quality of the responses, which cannot be evaluated by the compliance rate alone. The compliance rate is useful for assessing the direct effect of TSDI on the initial tokens, while the helpfulness win rate evaluates the overall quality of the model, which is important for practical use.
> >
> > > Q1 and W1: How does the "different type of vulnerability" mentioned by the authors differ from the conventional LLM safety vulnerabilities?
> >
> > A: We appreciate the reviewer's constructive comments. As the reviewer mentions, our initial statements in Section 2.2 would be misleading. The vulnerabilities this paper addresses are defined as the state of being weak for specific safety categories (i.e., adult content in our case) that safety-aligned LLMs mistakenly generate harmful responses. Such vulnerabilities have been studied in the research community (e.g., Wang et al (2023)), and it may be an exaggeration to call them "different" types of vulnerability. Based on the reviewer's comments, we have revised our paper. However, we still consider that it is a research contribution that 1) we find that safety-aligned LLMs trained using Safe RLHF or SACPO have serious vulnerability for specific categories (though they appear to be safe overall) and 2) we propose TSDI to mitigate them without further training.
> >
> > [1] Wang et al. "Do-not-answer: A dataset for evaluating safeguards in llms." arXiv preprint arXiv:2308.13387 (2023).
> >
> > > Q2: Is there evidence showing that TSDI is effective in "all safety categories," as the authors claim (I do not observe a significant improvement from Figure 1)?
> >
> > A: To provide evidence that the proposed method is effective for other categories, we conducted new experiments and now present new figures in Section 5 of our revised paper. Specifically, we provide new experiment results on two categories: 1) Trade and Compliance category and 2) Security Threats in addition to adult content. We think that we have provided sufficient empirical evidence to support that our TSDI works as a general algorithm to improve safety. We would like the reviewer to go through our experiment section, which we believe resolves the reviewer's concerns.

---

> > > ### Comment · Reviewer_c1NK · 2024-11-22
> > > **Thank you for the response**
> > >
> > > I appreciate the authors’ efforts in providing detailed responses to address my concerns. Hence, I would raise my score.

---

> > > > ### Author Response · Authors · 2024-11-22
> > > > **To Reviewer c1NK**
> > > >
> > > > Dear Reviewer,
> > > >
> > > > We sincerely appreciate the time and effort you spent on our work. Thank you!
> > > >
> > > > Best regards,
> > > >
> > > > Authors

---

### Official Review · Reviewer_VX4D · 2024-10-25

**Soundness:** 3
**Presentation:** 3
**Contribution:** 3
**Rating:** 6
**Confidence:** 4

**Summary:**

The paper discusses how to mitigate potential vulnerabilities in safe-aligned LLMs through TSDI, in order to better balance safety and helpfulness. The proposed method adjusts the model's output during the generation process to alleviate biases introduced by safety alignment. Experiment results show that TSDI significantly enhances helpfulness while ensuring safety, improving the trade-off between safety and helpfulness.

**Strengths:**

As a non-learning debiasing method, TSDI estimates bias through randomly generated token sequences and subtracts this bias during the generation process. This eliminates the need for additional training and complex computational resources, making it more cost-effective in practical applications, especially in resource-limited environments.

TSDI can improve the helpfulness of the model while maintaining safety. This addresses the common trade-off issue in current safety alignment methods, where typically an increase in safety leads to a decrease in helpfulness.

**Weaknesses:**

The proposed method only adjusts bias when generating the initial tokens and does not consider the model's hidden states. Some biases may arise from concepts in internal states, and token-level adjustments cannot fully mitigate them, so the effectiveness may be limited in more complex generation tasks. I suggest that the authors compare their work with those studying the role of safety in model hidden states [1, 2].

In contexts with lower helpfulness, the TSDI method can eliminate some negative tokens, but the overall improvement in helpfulness is not significant. Experimental results indicate that TSDI has not significantly improved the quality of responses in these contexts, making it insufficient for tasks that have a strong demand for helpfulness.

[1] https://arxiv.org/abs/2401.06824

[2] https://arxiv.org/abs/2404.12038

**Questions:**

In multi-category safety scenarios, how can you ensure that the debiasing effect of TSDI is applicable to all categories?

---

> ### Author Response · Authors · 2024-11-19
> **Authors' replies to Reviewer VX4D**
>
> We deeply appreciate the reviewer’s encouraging comments. The thoughtful comments of Reviewer VX4D are valuable and help us improve the manuscript's quality. We will answer the comments in Weaknesses and Questions.
>
> > W1: The proposed method only adjusts bias when generating the initial tokens and does not consider the model's hidden states. Some biases may arise from concepts in internal states, and token-level adjustments cannot fully mitigate them, so the effectiveness may be limited in more complex generation tasks. I suggest that the authors compare their work with those studying the role of safety in model hidden states [1, 2]. [1] https://arxiv.org/abs/2401.06824 [2] https://arxiv.org/abs/2404.12038
>
> A: The existing methods the reviewer mentions are on adversarial attacks that utilize hidden states; hence, we consider that it is inappropriate to compare our TSDI with them. A major advantage of TSDI is that it can be used as long as we can access logits. That said, we agree with the reviewer that References [1] and [2] are strongly related to our method; hence, we have added them as references in Section 5.2 in our revised version.
>
> > W2: In contexts with lower helpfulness, the TSDI method can eliminate some negative tokens, but the overall improvement in helpfulness is not significant. Experimental results indicate that TSDI has not significantly improved the quality of responses in these contexts, making it insufficient for tasks that have a strong demand for helpfulness.
>
> A: The primary objective of TSDI is to improve the safety of the existing safety-aligned LLMs. In this process, the TSDI must ensure that helpfulness does not significantly deteriorate as a requirement. As shown in Figure 1, TSDI achieves better balances between helpfulness and safety, which demonstrates that TSDI successfully improves the safety of a safety-aligned LLM (by SACPO) while maintaining its helpfulness.
>
> > Q: In multi-category safety scenarios, how can you ensure that the debiasing effect of TSDI is applicable to all categories?
>
> A: The debiasing effect of TSDI is applicable to multi-category safety scenarios because TSDI is a general algorithm that is not specific to any category. In particular, TSDI's bias estimation and mitigation procedure does not require any information about the categories. To show the effect of debiasing on other categories, we added new experiments and now present new figures in Section 5 of our revised paper. Specifically, we conducted new experiments on two categories: 1) Trade and Compliance category and 2) Security Threats in addition to adult content. We would like the reviewer to go through our experiment section, which we believe resolves the reviewer's concerns.

---

> ### Comment · Reviewer_VX4D · 2024-11-19
>
> I thank the authors for clarifying my concerns. Based on this, I raise the confidence of the score.

---

> > ### Author Response · Authors · 2024-11-21
> > **To Reviewer VX4D**
> >
> > Dear Reviewer,
> >
> > We sincerely appreciate the time and effort you spent on our work. Thank you!
> >
> > Best regards,
> >
> > Authors

---

### Official Review · Reviewer_mi6Q · 2024-10-30

**Soundness:** 3
**Presentation:** 2
**Contribution:** 2
**Rating:** 5
**Confidence:** 4

**Summary:**

This paper discusses the safety vulnerabilities of LLMs in specific categories. Even when the models perform well in the general sense of safety, the models may suffer in specific safety categories. The authors developed a token-wise debiasing approach to enhance the safety performance.

**Strengths:**

The paper discusses the safety of LLMs in specific categories. Both theoretical and experimental results are presented to support the paper claims.

**Weaknesses:**

I appreciate the efforts and experiments performed by the authors. But the findings presented in Section 3 have been known to the community, at least qualitatively. The experiments lack baseline comparisons and hence are not convincing. Please see questions for more details.

**Questions:**

- From Fig. 1, it is hard to see how TSDI improves compared to beaver v3 and SACPO, despite the numbers shown in legend. It would be great to see some statistical significance of the proposed approach.
- The results in Fig. 1 seem to indicate that all methods struggle with category 3, including TSDI. I wonder if these results are sufficient to support the motivation of this paper. The authors could provide more details to better support the statements or motivation of the paper.
- I wonder whether it is meaningful to use SFT as the comparison baseline. The primary goal of SFT is not safety, and using it as a baseline in Fig.1 and later experiments seems unfair.
- From Fig. 6, it is hard to see how significant TSDI is compared to situations without TSDI. Some statistical significance should be presented to better support the claim.
- There is no safety baselines being evaluated in this paper and compared with TSDI. The authors should consider adding safety baselines, such as SmoothLLM, Llama guard, in-context demonstration, self-reminder, safe decoding, etc, to demonstrate the superior of TSDI
- How would one get a reference model? Is equation 7 applied to all steps $i$ or not? It would be great to see some ablation study on the choice of reference models.
- What are the difference between the points in Fig. 5? Are they generated using different hyper-parameters?
- I may have missed this. The authors showed that cleaning data may not help. How about increase the data quantity, say in the adult content category? Some experimental analysis along this direction may better support this paper.

---

> ### Author Response · Authors · 2024-11-19
> **Authors' replies to Reviewer mi6Q (1/2)**
>
> We appreciate the valuable feedback from the reviewer. A revision paper has been uploaded. We will first address the comments in Weaknesses and then answer the Questions.
>
> > W: But the findings presented in Section 3 have been known to the community, at least qualitatively.
>
> A: We respectfully disagree with the reviewer saying that our findings are known to the community. Recent safety alignment papers (e.g., SACPO and CAN, both just accepted at NeurIPS-24) do not discuss such vulnerability and claim the safety of their resulting models; thus, we believe that the safety alignment community is not necessarily aware of this issue. As the safety alignment research has grown extremely fast, we may miss some important papers. We welcome specific and concrete citation suggestions directly relevant to our paper from reviewers with close expertise. Based on the suggestions, we are happy to resolve the reviewer's concerns and revise our paper during this discussion period.
>
> > Q1: From Fig. 1, it is hard to see how TSDI improves compared to beaver v3 and SACPO, despite the numbers shown in legend. It would be great to see some statistical significance of the proposed approach.
>
> A: We deeply thank the reviewer for the helpful comment. We have improved the readability of Figure 1 and believe that it can resolve the reviewer's concern. Compared to SACPO, TSDI achieved a higher safety score in categories 3, 6, 9, 10, and 12, and the mean safety score is also higher. Since beaver-7b-v2.0 and -v3.0 achieved a similar safety score to TSDI, their helpfulness is lower than ours. We also emphasize that beaver-7b-v2.0 and -v3.0 are trained with a much larger dataset and more training iterations than our setting. We also conducted statistical significance testing of the proposed method in Appendix C.8 (Please see Q4 for more details.)
>
> > Q2: The results in Fig. 1 seem to indicate that all methods struggle with category 3, including TSDI. I wonder if these results are sufficient to support the motivation of this paper. The authors could provide more details to better support the statements or motivation of the paper.
>
> A: We understand the reviewer's concerns about our proposed method, which may seem to struggle with category 3. However, we would like to emphasize that our TSDI still achieves the highest level of safety. We consider that it is of practical importance to deploy models that are as safe as possible for all safety categories. Also, TSDI well balances helpfulness and safety, unlike beaver-7b-v2.0 and -v3.0, which have low levels of helpfulness. Regarding the safety level, our TSDI obtains a very high safety score when evaluated by the Llama Guard 3 (see Figure 12). Although the safety score by MD-Judge may seem to be low, please note that MD-Judge tends to give a severe assessment for adult-related sentences (i.e., category 3).
>
> > Q3: I wonder whether it is meaningful to use SFT as the comparison baseline. The primary goal of SFT is not safety, and using it as a baseline in Fig.1 and later experiments seems unfair.
>
> A: We do *not* regard the SFT model as the baseline for evaluating safety. We rather claim that such a win rate is not an appropriate metric for safety. This is exactly the reason why we use MD-Judge and Llama Guard 3 to assess the safety performance of each model (i.e., the y-axis of Figure 1). We would like to emphasize that we used the SFT models as the baseline for evaluating helpfulness.

---

> > ### Author Response · Authors · 2024-11-19
> > **Authors' replies to Reviewer mi6Q (2/2)**
> >
> > > Q4: From Fig. 6, it is hard to see how significant TSDI is compared to situations without TSDI. Some statistical significance should be presented to better support the claim.
> >
> > A: We thank the reviewer for the valuable comment. We have added significance testing in Appendix C.8, which greatly supports our claim and addresses the reviewer's concern. In particular, we adopt hypervolume as the metric to compare two Pareto fronts (see Appendix C.8 for more details). We conducted TSDI with three different random seeds and computed the hypervolume of the Pareto front between the helpfulness win rate and safety score. Finally, we obtained the mean and standard deviation of the hypervolume across the three runs and observed that the improvement by TSDI is statistically reliable.
> >
> > > Q5: There is no safety baselines being evaluated in this paper and compared with TSDI. The authors should consider adding safety baselines, such as SmoothLLM, Llama guard, in-context demonstration, self-reminder, safe decoding, etc, to demonstrate the superiority of TSDI
> >
> > A: We deeply thank the reviewer for the valuable comment. We agree that a comparison with these safety baselines can further strengthen our paper's claim. Consequently, we have added a new experiment in Appendix B.7 to compare TSDI with Self-reminder and SafeDecoding. We chose Self-reminder due to its simplicity and widespread adoption in practice. On the other hand, we selected SafeDecoding because its framework is very similar to TSDI, as the reward-aligned model and safety-aligned model in TSDI can be considered analogous to the base model and expert model in SafeDecoding, respectively. Our experimental results showed that TSDI achieved a better safety-helpfulness trade-off Pareto front compared to SafeDecoding and Self-reminder. While Self-reminder can improve the safety of the models, it does not neccesarily maintain the helpfulness of the model. Conversely, SafeDecoding maintains the helpfulness of the model but fails to achieve a high safety level. These results highlight the challenges of our problem and the effectiveness of TSDI.
> >
> > > Q6: How would one get a reference model? Is equation 7 applied to all steps or not? It would be great to see some ablation study on the choice of reference models.
> >
> > A: We consider the safety alignment problem in which the SFT model is readily available. In our paper, the reference model is obtained by aligning the SFT model for reward. Since we intend to remove the unwanted effects of safety alignment, Equation (7) is applied after the second step, where safety alignment is conducted.
> >
> > > Q7: What are the difference between the points in Fig. 5? Are they generated using different hyper-parameters?
> >
> > A: Yes, there are generated using different $\beta/\lambda$ and training iterations.
> >
> > > Q8: I may have missed this. The authors showed that cleaning data may not help. How about increase the data quantity, say in the adult content category? Some experimental analysis along this direction may better support this paper.
> >
> > A: For the safe of (safety) alignment, we need a large amount of high-quality data, and the data collection process typically depends on humans. Increasing the amount of data is costly, and there are many options for how to increase the quantity. We agree with the reviewer that increasing the data quantity can be a powerful solution, so we will leave it for future work.

---

> > > ### Comment · Reviewer_mi6Q · 2024-11-20
> > >
> > > I appreciate the authors' effort and time in addressing the comments. After going through the revision, I still have the following questions.
> > >
> > > I may misunderstand Section 3. However, existing works on benchmarking LLM safety such as HarmBench have shown that existing models are not superior in all categories. Although the categories may be different, I do not see how Section 3 leads to new results than existing studies.
> > >
> > > Correct me if I am wrong. A t-test may be more convincing than the current results in Appendix C.8.
> > >
> > > After reading through the authors' response, I am more confused on why SFT is used in Figure 1 (left panel). If SFT is not intended for comparison on safety, it should not be used in Figure 1 whose y-axis is safety score. This could be misleading to readers.
> > >
> > > A minor comment: The value of n is missing in line 1412.

---

> > > > ### Author Response · Authors · 2024-11-21
> > > > **Authors' replies to Reviewer mi6Q**
> > > >
> > > > We thank the reviewer for taking the time and effort to review our paper. We have carefully considered the reviewer's comments and submitted a revised version addressing the feedback provided.
> > > >
> > > > > Q1: I may misunderstand Section 3. However, existing works on benchmarking LLM safety, such as HarmBench, have shown that existing models are not good in all categories. Although the categories may be different, I do not see how Section 3 leads to new results compared to existing studies.
> > > >
> > > > Thank you for your valuable feedback and for mentioning HarmBench. We acknowledge the close relationship between our work and existing studies, and we have added the reference to HarmBench and updated our discussion to the related works in Section 2.2. However, we believe that Section 3.1 still offers unique research contributions by highlighting specific issues in recent safety alignment papers, such as SafeRLHF and SACPO (both just accepted at NeurIPS-24). Additionally, in Sections 3.2 and 3.3, we conducted large-scale experiments to explore the challenges of simultaneously achieving high safety and helpfulness. We believe these findings contribute to the community and offer new insights beyond existing studies.
> > > >
> > > > > Q2: Correct me if I am wrong. A t-test may be more convincing than the current results in Appendix C.8.
> > > >
> > > > Thank you for your comment. We agree that a t-test can be a valuable tool for statistical analysis. However, in our situation, the difference between the two methods is significantly larger than the standard deviation, which strongly suggests that our method is statistically superior. Although conducting a t-test is not essential in this context, we provide the t-test results as addition evidence to reinforce our conclusion.
> > > >
> > > > ### t-testing results for cases using MD-Judge as the safety evaluator
> > > >
> > > > | Category                  | p-value  |
> > > > |---------------------------|----------|
> > > > | O3: Adult Content         | 0.0083   |
> > > > | O7: Trade and Compliance  | 0.0018   |
> > > > | O10: Security Threats     | 0.0029   |
> > > >
> > > > ### t-testing results for cases using Llama Guard as the safety evaluator
> > > >
> > > > | Category                  | p-value  |
> > > > |---------------------------|----------|
> > > > | O3: Adult Content         | 0.0030   |
> > > > | O7: Trade and Compliance  | 0.0032   |
> > > > | O10: Security Threats     | 0.0034   |
> > > >
> > > > > Q3: After reading through the authors' response, I am more confused on why SFT is used in Figure 1 (left panel). If SFT is not intended for comparison on safety, it should not be used in Figure 1 whose y-axis is safety score. This could be misleading to readers.
> > > >
> > > > Thank you for your insightful comment. We first emphasize that we have explicitly stated that the safety of the target model should not be assessed through pair-wise comparisons with the SFT model (Lines 55-57). In particular,
> > > > - SafeRLHF, SACPO, and CAN evaluate safety by comparing the win rate against SFT.
> > > > - We employ safety evaluators (MD-Judge and LLaMA Guard 3) to compute safety scores, which we then compare with the scores of SFT or other methods.
> > > >
> > > > Since SFT exhibits very low safety performance, a model evaluated solely using safety win-rate versus the SFT model might be deemed safe, even if it is not truly safe. Moreover, the SFT model is included in Figure 1 for two main reasons.
> > > > 1. To demonstrate that while the existing method enhances the overall safety score of the SFT model, it still falls short in certain specific categories (see Lines 237-238).
> > > > 2. To highlight the challenge of predicting categories with low safety scores solely by examining the SFT safety scores (see Lines 296-298).
> > > >
> > > > That said, we have revised the discussion to improve clarity on these points.
> > > >
> > > > > Q4: The value of n is missing in line 1412.
> > > >
> > > > Thank you for your comment. We have revised this part.

---

> > > > > ### Comment · Reviewer_mi6Q · 2024-11-22
> > > > > **Thank you and I will raise the score**
> > > > >
> > > > > I appreciate the authors for performing additional experiments and addressing my comments. As acknowledged by the authors, I am still not fully convinced by the novelty of conclusions in Section 3. But I will raise my score for the new experiments on statistical significance and baseline comparisons.

---

> > > > > > ### Author Response · Authors · 2024-11-22
> > > > > > **To Reviewer mi6Q**
> > > > > >
> > > > > > Dear Reviewer,
> > > > > >
> > > > > > We sincerely appreciate the time and effort you spent on our work. Thank you!
> > > > > >
> > > > > > Best regards,
> > > > > >
> > > > > > Authors

---

### Official Review · Reviewer_Tfzj · 2024-11-04

**Soundness:** 2
**Presentation:** 3
**Contribution:** 3
**Rating:** 6
**Confidence:** 3

**Summary:**

The paper is basically based on alignment of LLMs in such a way where helpfulness does not get compromised during safety tuning of LLMs. The authors proposed a method which they have coined as TSDI ensures that the bias which originates in the initial tokens of the output while Safety Tuning an LLM is taken care of. Their concern is that these unwanted biases reduce the helpfulness of the Models after safety tuning regardless of the training parameters and quality dataset used.

**Strengths:**

1. The approach is quite innovative and the method introduced is quite novel.
2. The in-depth portrayal of the previous works and its relation to their new work is quite in sync and properly depicted in the paper
3. Analysing all the existing parameters, training setup, methodology and dataset used in order to conclude why their method is unique and effective is commendable.

**Weaknesses:**

1.	Line no 54, “Previous studies on safety alignment suffer from misdirected safety evaluation”. Please add reference.
2.	In Figure 1 as well as line no 235, what does the term “Helpfulness win rate” refers to? I didn’t find its definition or explanation of what it is or how it is being calculated.
3.	In line no 257, beta = 0.1 chosen. Is there any significance of choosing 0.1 or it was a random start?
4.	In Figure 2 (a), I can see that with decreasing (beta/lamda), the safety score for adult category is increasing with number of iterations. But in line 266, it is written with decreasing (lamda/beta), the safety scores increase. There seems to be an ambiguity here.
5.	In line 267, it is written “We also observed that a small (beta/lamda) and excessive training iterations sometimes led to generation corruption”. Please provide some supporting documents in this regard.
6.	In line 309, can you please provide the analysis of changing 0.25 (0.1, 0.2, 0.3, 0.4) Like how the safety scores and Helpfulness of the model changes. It would be great to know how much is the drop in the Helpfulness while an increase in the Safety Score. There can be instances where by removing a small number of datapoints from the actual data, the safety score increases more as compared to the dip in Helpfulness. I want to know this possibility.
7.	In Figure 4, in the legend, what does the numbers in the square bracket represents?
8.	In Figure 4, 1st Graph, the x axis should be (beta/lamda) with respect to what it has been described. Are you changing the Beta in this case? So, is both Beta and Lamda changing or you are keeping beta fixed and changing Lamda to analyse the output?
9.	In line 364, why only MMLU dataset has been chosen for generating random prompts and response? Can you do it with some other dataset just to check consistency over other dataset distribution as well.
10.	Can you please clarify this? You claimed that TSDI is a learning free method to estimate and mitigate the bias introduced by the safety alignment. (Line 387) But in order to get the Biases, you are subtracting the output probabilities of initial output tokens of a Safety Aligned Model and a Reference Model. How are you getting that Safety Aligned Model without learning or safety tuning?
11.	Can you please show the variation of Safety Score and Compliance Rate with respect to the change in L (Initial number of tokens taken into consideration for debiasing)?
12.	In figure 5, why for a particular dataset, we are getting a lot of points corresponding to different Compliance Rate and Adult Safety Score? Are those different points referring to different settings (Different beta/ lamda value or different number of iterations).
13.	In Figure 6, similar confusion as mentioned in Point 12
14.	Just to check that the general capabilities of LLMs have not been hampered, can you show some benchmarking results based on your approach. (For example. MMLU, HellaSwag, ARC)

**Questions:**

There are quite a few things that need to be taken care to make the paper more impactful and understandable. Please look into the weakness section and address the issues.

---

> ### Author Response · Authors · 2024-11-19
> **Authors' replies to Reviewer Tfzj (1/2)**
>
> We appreciate the valuable feedback from the reviewer. A revision paper has been uploaded. Below is our response to the comments in Weaknesses.
>
> > W1: Line no 54, “Previous studies on safety alignment suffer from misdirected safety evaluation”. Please add reference.
>
> A: Thank you for your helpful comments! We have added references in our revised paper.
>
> > W2: In Figure 1 as well as line no 235, what does the term “Helpfulness win rate” refers to? I didn’t find its definition or explanation of what it is or how it is being calculated.
>
> A: Helpfulness win rate represents the percentage that the model to be evaluated can successfully generate more helpful responses than the SFT model. We have added explanations of this metric in Section 3.1.
>
> > W3: In line no 257, beta = 0.1 chosen. Is there any significance of choosing 0.1 or it was a random start?
>
> A: In RLHF or DPO, $\beta=0.1$ is a common parameter as the reverse KL penalty coefficient. We also chose $\beta=0.1$ as a typical value.
>
> > W4: In Figure 2 (a), I can see that with decreasing (beta/lamda), the safety score for adult category is increasing with number of training iterations. But in line 266, it is written with decreasing (lamda/beta), the safety scores increase. There seems to be an ambiguity here.
>
> A: We apologize for our typo. We have corrected it in our revised version.
>
> > W5: In line 267, it is written “We also observed that a small (beta/lamda) and excessive training iterations sometimes led to generation corruption”. Please provide some supporting documents in this regard.
>
> A: In the cases of a small $\beta/\lambda$ and excessive training iterations, the resulting aligned model will deviate significantly from the reference model. Consequently, the model sometimes outputs a large number of Greek letters and Russian words, which would be unobservable in a normal model. We also added some examples for corruption in Appendix D.3.
>
> > W6: In line 309, can you please provide the analysis of changing 0.25 (0.1, 0.2, 0.3, 0.4) Like how the safety scores and Helpfulness of the model changes. It would be great to know how much is the drop in the Helpfulness while an increase in the Safety Score. There can be instances where by removing a small number of datapoints from the actual data, the safety score increases more as compared to the dip in Helpfulness. I want to know this possibility.
>
> A: After trying various thresholds, we concluded that a threshold of 0.25 was the optimal parameter that improved the safety score. Our interpretation is that a large threshold did not have a sufficient cleansing effect because a small number of data was removed. On the other hand, when a smaller threshold was used, the safety alignment did not work well because useful data was also removed. This analysis of changing the threshold requires a significant amount of training time, which we are unable to conduct within this short discussion period. However, we believe this is an interesting problem and plan to explore more in future work.
>
> > W7: In Figure 4, in the legend, what does the numbers in the square bracket represents?
>
> A: The numbers in the square brackets represent the tokens in each token group. For example, the tokens used in the group `none` are `5642, 6213, 8516, 9290`, which are the encoding of `none, None, None, none`, respectively. A better explanation should have been clearly stated in our paper, hence, we have added the decoded text of these tokens in Appendix C.7 and updated the captions of Figures 4 and 9.
>
> > W8: In Figure 4, 1st Graph, the x axis should be (beta/lamda) with respect to what it has been described. Are you changing the Beta in this case? So, is both Beta and Lamda changing or you are keeping beta fixed and changing Lamda to analyse the output?
>
> A: We apologize for our typo. This $\beta$ should be $\beta/\lambda$ in Figure 4. We have revised our paper.
>
> > W9: In line 364, why only MMLU dataset has been chosen for generating random prompts and response? Can you do it with some other dataset just to check consistency over other dataset distribution as well.
>
> A: Thank you for your insightful comments! We added additional experiments in Appendix B.6. Our new experiments show that our proposed method exhibits a stable and consistent performance regardless of dataset choices.

---

> > ### Author Response · Authors · 2024-11-19
> > **Authors' replies to Reviewer Tfzj (2/2)**
> >
> > > W10: Can you please clarify this? You claimed that TSDI is a learning free method to estimate and mitigate the bias introduced by the safety alignment. (Line 387) But in order to get the Biases, you are subtracting the output probabilities of initial output tokens of a Safety Aligned Model and a Reference Model. How are you getting that Safety Aligned Model without learning or safety tuning?
> >
> > A: We call our bias estimation and mitigation algorithm (i.e., TSDI) learning-free, where the baseline safety-aligned model is given. We emphasize as a key contribution that our TSDI can improve the safety of the existing safety-aligned model without further training.
> >
> > > W11: Can you please show the variation of Safety Score and Compliance Rate with respect to the change in L (Initial number of tokens taken into consideration for debiasing)?
> >
> > A: We appreciate the reviewer's helpful suggestions! We added new experimental results in Appendix B.5. The new empirical results show that our TSDI is robust against how to choose $L$.
> >
> > > W12: In figure 5, why for a particular dataset, we are getting a lot of points corresponding to different Compliance Rate and Adult Safety Score? Are those different points referring to different settings (Different beta/ lamda value or different number of iterations.
> >
> > > W13: In Figure 6, similar confusion as mentioned in Point 12
> >
> > A: Yes, the different points correspond to the combinations of different $\beta/\lambda$ and the number of iterations. We should have clearly stated this in our paper; hence, we have updated the captions of Figures 5 and 6.
> >
> > > W14: Just to check that the general capabilities of LLMs have not been hampered, can you show some benchmarking results based on your approach. (For example. MMLU, HellaSwag, ARC)
> >
> > A: Evaluation by GPT-4 is a popular and standard approach in LLM alignment research. Existing studies have shown that GPT-4 is well-aligned with and is a good proxy for human evaluation. For example, in Section 6.4 of the DPO paper (NeurIPS version), they mentioned that “GPT-4 tends to agree with humans about as often as humans agree with each other, suggesting that GPT-4 is a reasonable proxy for human evaluations.” There is a similar discussion in Section 4.2.1 of the Safe-RLHF paper: “When compared to Alpaca-7B, the Beaver-v3 model demonstrated an increase in the Elo score for helpfulness (GPT-4: +244.91, Human: +363.86) and for harmlessness (GPT-4: +268.31, Human: +237.98). Comparatively, the evaluations by GPT-4 and human evaluators are almost consistent.” Recent safety alignment papers (e.g., SACPO and CAN) are also based on this LLM-as-a-Judge style evaluation by GPT-4. Therefore, we consider that our evaluation is based on a de facto standard for evaluating the general capability of LLMs.

---

> > > ### Comment · Reviewer_Tfzj · 2024-11-21
> > > **Official Comment by Reviewer Tfzj**
> > >
> > > Thank you for addressing the issues I had raised. I have got quite a good understanding of your paper. But this paper contains too many revisions in various sections. It needs a thorough reconstruction in order to properly address those issues and require some restructuring of the contents.
> > > And to the best of my knowledge, it is not recommended to resubmit the revised version of the paper after the reviews are rolled out. Taking all these into consideration, I would like to keep my scores same.

---

> > > > ### Comment · Area_Chair_FYEr · 2024-11-22
> > > >
> > > > Dear reviewer Tfzj,
> > > >
> > > > I would like to clarify a point.
> > > >
> > > > Revising the paper (adding experiments, references, explanations, comparisons, etc) based on the reviewers' feedback is not only allowed but actually encouraged, rather than promising revisions in the camera ready.
> > > >
> > > > Having said that, if the revisions turn out to be so major that the paper is effectively becomes a new paper, then the reviewers can reserve the right to not consider those revisions.
> > > >
> > > > Thanks,\
> > > > AC

---

> > > > > ### Author Response · Authors · 2024-11-22
> > > > > **To Area Chair FYEr**
> > > > >
> > > > > Dear Area Chair FYEr,
> > > > >
> > > > > We deeply thank you for clarifying the review process guidelines with the reviewer.
> > > > >
> > > > > Best regards,
> > > > >
> > > > > Authors

---

> ### Author Response · Authors · 2024-11-21
> **Authors' replies to Reviewer Tfzj**
>
> We thank the reviewer for taking the time and effort to review our paper. In our humble opinion, we would like to argue that submitting revised versions during the discussion phase is explicitly allowed and even encouraged according to the ICLR Author and Reviewer Guidelines. We kindly ask the reviewer to refer to the guidelines again, if possible, for a fair review process. Additionally, we want to emphasize that the revised version of our paper is well-structured, and the changes made do not compromise the overall structure and quality. We believe these revisions enhance the clarity and depth of our work.

---

### Official Review · Reviewer_zvdo · 2024-11-11

**Soundness:** 2
**Presentation:** 3
**Contribution:** 3
**Rating:** 6
**Confidence:** 3

**Summary:**

The paper exposes and reiterates some challenges with achieving a good safety-helpfulness tradeoff in safety alignment and proposes a token-level method to debias safety alignment to achieve the same.

**Strengths:**

1) The paper is really well-written. Despite not working in the area actively, I could easily understand the paper. The graphs and captions are very clear.
2) I found the challenges in improving dataset section to be very enlightening. Interesting to know that a category can have not low quality or quantity and still suffer in safety score. Also interesting to know that data cleaning did not bring many benefits here.
3) The paper is timely and the topic is interesting + relevant.

**Weaknesses:**

1) I think the biggest weakness of the paper is its very limited experimental section. It would help to explore atleast other categories. Why not also include other categories with a similar behavior and quality-quantity distribution? And explore the effect of debiasing on other categories with a different behavior and quality-quantity distribution?
2) As it seems from the results, the method doesn't give significant gains on helpful win rate which we actually care about. Are there cases where your token-level method might still be beneficial? maybe where compliance and helpful win rates are highly similar?
3) While it is good to reiterate and provide more evidence for, I am not sure that the first two challenges are novel.

**Questions:**

1) how & why do you choose 0.25 threshold for cleaning the data?

---

> ### Author Response · Authors · 2024-11-19
> **Authors' replies to Reviewer zvdo**
>
> We deeply appreciate the reviewer zvdo for helpful and thoughtful comments and questions. A revision paper has been uploaded. We will first address the comments in Weaknesses and then answer the Question.
>
> > W1: I think the biggest weakness of the paper is its very limited experimental section. It would help to explore atleast other categories. Why not also include other categories with a similar behavior and quality-quantity distribution? And explore the effect of debiasing on other categories with a different behavior and quality-quantity distribution?
>
> A: We deeply thank the review for the comment. The reason why we focused on adult content (i.e., category 3) is that our target models (by SafeRLHF and SACPO) had serious vulnerability on the topic. However, our proposed TSDI is not specific to this adult content category, and we consider it a general algorithm. As evidence, we conducted new experiments and now present new figures in Section 5 of our revised paper. Specifically, we provide new experiment results on two categories: 1) Trade and Compliance category and 2) Security Threats in addition to adult content. We would like the reviewer to go through our experiment section (especially Figures 5 and 6), which we believe resolves the reviewer's concerns.
>
> > W2: As it seems from the results, the method doesn't give significant gains on helpful win rate which we actually care about. Are there cases where your token-level method might still be beneficial? maybe where compliance and helpful win rates are highly similar?
>
> A: We thank the review for the comment. The primary objective of our TSDI is to improve the safety of existing safety-aligned LLMs while maintaining their helpfulness. In other words, we do not aim to merely improve the helpfulness of the model in our paper. Compared to compliance rate, the helpful win rate did not improved as the same extend. We recognized this as a challenge and will explore for alternative solutions in future work. Still, as we additionally show in Figure 1 in our revised version, our TSDI well balances helpfulness and safety compared to the baseline methods.
>
> > W3: While it is good to reiterate and provide more evidence for, I am not sure that the first two challenges are novel.
>
> A: We believe that we have cited the most relevant papers at the time of the ICLR 2025 submission. We welcome specific and concrete citation suggestions directly relevant to our paper from reviewers with close expertise. Based on the suggestions, we are happy to revise our paper during this discussion period.
>
> > Q: how \& why do you choose 0.25 threshold for cleaning the data?
>
> A: After trying various thresholds, we concluded that a threshold of 0.25 was the optimal parameter to improve the safety score. Our interpretation is that a large threshold did not have a sufficient cleansing effect because a small number of data was removed. On the other hand, when a smaller threshold was used, the safety alignment did not work well because useful data was also removed.

---

### Author Response · Authors · 2024-11-19
**To all reviewers**

Dear Respected Reviewers,

We sincerely appreciate the time and effort the reviewers have dedicated to reviewing our paper. Your valuable feedback has been instrumental in improving our work, and we have submitted the revised version to OpenReview.

Below is a brief overview of the major modifications made:

- **Experiment results on other categories:** To demonstrate the effectiveness of TSDI across different categories, we have added new experimental results in Section 5 for two additional categories: Trade and Compliance, and Security Threats, in addition to Adult Content. (In response to reviewers zvdo, VX4D, and c1NK)

- **Improved Figure 1:** We have reconstructed Figure 1 to enhance its readability, highlighting TSDI's efficiency in improving safety while maintaining the model's helpfulness. (In response to reviewers zvdo, mi6Q, VX4D, and c1NK)

- **Experiment results on robustness:** We have included new experiments in Appendix B.5 and B.6 to show TSDI's robustness to the choice of $L$ (the length of the random prompt) and the dataset used for constructing the token pool, respectively. (In response to reviewer Tfzj)

- **Comparison with safety baselines:** We added a new experiment in Appendix B.7 comparing TSDI with two safety baselines: SafeDecoding and Self-reminder. (In response to reviewer mi6Q)

- **Significance testing:** We conducted significance testing in Appendix C.8 to demonstrate that the improvements achieved by TSDI are statistically reliable. (In response to reviewer mi6Q)

- **Updated discussion on vulnerabilities of safety-aligned models:** We revised Section 2.2, adding new references and updating the discussion on the relation to related works. (In response to reviewer c1NK)

- **Addressed missing citations:** We have thoroughly addressed your suggestions regarding missing references. In Sections 1 and 5.2, we discussed and cited the papers mentioned by the reviewers. (In response to reviewers Tfzj and VX4D)

- **Provided missing explanations:** We clarified the helpfulness win rate in Section 3.1 (In response to reviewer Tfzj), detailed the difference of points in Figures 5 and 6 in their captions (In response to reviewers Tfzj and mi6Q), and explained the meaning of the numbers in brackets in Figure 4 (In response to reviewer Tfzj).

The changes in the paper are highlighted using red text. Once again, we genuinely appreciate your dedication to the review process and look forward to your continued insights.

Best Regards,

Authors.

---

### Meta-Review · Area_Chair_FYEr · 2024-12-23

**Metareview:**

This paper explores a fundamental issue in safety alignment: when a model is aligned for safety, the refusal rate increases even on benign queries. The authors explore this issue and provide a compelling analysis on understanding the issue in detail. As mentioned by the reviewers, there are other works that try to highlight a similar issue, and the authors connected with some of those works in their rebuttal. The main finding is that the likelihood of some refusal tokens increases significantly in the first few tokens of decoding. Given the insights, the proposal here is to subtract the average of those logits at decoding time to provide an inference-time solution to debias the model. The experiments show that this method results in a Pareto frontier that slightly dominates those of the existing models, which is a proof-of-concept built on the insights. Overall, this paper is well written and well executed but still needs to be improved to better connect with the literature. First, the paper is effectively proposing an inference-time alignment method. This needs to be better understood as compared to other existing techniques such as those of (Deng and Raffel, 2023) and (Mudgal et al., 2023). Additionally, the reviewers mentioned comparisons between token-wise interventions and those in embedding space could be an additional area for investigation, see (Zou et al., 2024). There is also related work with similar understanding of the shallowness of safety alignment focusing on the first few tokens (Qi et al., 2024). Given their findings, it is also important to understand what the effect of the proposal on adversarial jailbreaks is as intuitively the model may become even more susceptible given the removal of the bias on subsequent token positions. Finally, the AC also found a fundamental issue with Proposition 1 (and its proof), which is detailed below and needs to be fixed in the next iteration of the paper. We hope that the authors find these comments helpful for revising their paper and submitting to the next venue.

Issue with Proposition 1: The authors assume that the safety reward acts token-wise on any prefix of the generation. This is unfortunately not the case as the safety reward can only identify whether a fully decoded response is safe. Further, the first equation in the proof identifying the token-probability in Ln 762 does not hold. However, this could be fixed by replacing the safety reward with a value function for safety and then the Eq. in Ln 762 would be an instance of (Mudgal et. al. 2023, Theorem 2.1) when considering a token-wise reinforcement learning problem, different from what is assumed here. In fact, connecting the decoding method with the token-wise MDP defined by Mudgal et. al. (2023) would be useful beyond fixing Proposition 1.

Deng, Haikang, and Colin Raffel. "Reward-augmented decoding: Efficient controlled text generation with a unidirectional reward model." arXiv preprint arXiv:2310.09520 (2023).

Mudgal, Sidharth, et al. "Controlled decoding from language models." arXiv preprint arXiv:2310.17022 (2023).

Zou, Andy, et al. "Improving Alignment and Robustness with Short Circuiting." arXiv preprint arXiv:2406.04313 (2024).

Qi, Xiangyu, et al. "Safety Alignment Should Be Made More Than Just a Few Tokens Deep." arXiv preprint arXiv:2406.05946 (2024).

**Additional Comments On Reviewer Discussion:**

All reviewers think that the paper is interesting and appreciated the proposal but felt that it falls short of acceptance. In addition, the AC found an issue with Proposition 1 and its proof that needs to be fixed in a next iteration of the paper.

---

### Decision · Program_Chairs · 2025-01-22

Reject